# Body Composition versus BMI as Measures of Success in a Clinical Pediatric Weight Management Program

**Kristin Stackpole [1], Philip Khoury [1], Robert Siegel [1,2,\*] and Amanda Gier [1]**

1    Cincinnati Children's Hospital Medical Center, Cincinnati, OH 45229, USA;
     kristin.stackpole@cchmc.org (K.S.); phil.khoury@cchmc.org (P.K.); Amanda.gier@cchmc.org (A.G.)
2    School of Medicine, University of Cincinnati, Cincinnati, OH 45221, USA
\*    Correspondence: bob.siegel@cchmc.org

**Abstract:** The high rates and long-term medical consequences of childhood obesity make it a public health crisis requiring effective diagnosis, treatment, and prevention. Although BMI is an adequate screening tool for obesity, monitoring BMI change is not always the best measure of success in treating patients in a pediatric weight management program. Our retrospective study evaluated the proportion of patients that achieved favorable changes in body composition by bioelectrical impedance analysis in the absence of improvements in BMI, BMI percentile, or percent of the 95th percentile for BMI. It was found that 30% of patients whose BMI increased by 1.0 kg/m$^2$ or more, 31.6% of patients with stable or increasing BMI percentiles, and 28% with stable or increasing percent of the 95th percentile for BMI demonstrated an improvement in body composition (skeletal muscle mass and body fat percentage). Body composition is an important measure of success for a subset of patients who otherwise may believe that their efforts in lifestyle change have not been effective. Our results suggest that including body fat percentage as a measure of success in evaluating the progress of patients participating in a pediatric weight management program is appropriate and may more accurately track success than change in BMI or BMI percentile alone.

**Keywords:** childhood obesity; body composition; pediatric weight management; percent body fat

---

## 1. Introduction

Childhood obesity is a public health crisis. The rate of obesity among children living in the United States age 2 to 19 years old is 18.5% and 6.0% of US youth are classified as severely obese [1]. Rates of obesity and severe obesity in adolescents have doubled and quadrupled respectively over the past 30 years. The rate of obesity among adolescents, age 12–19 years old, increased from 10.5% in 1988–1994 to 20.6% in 2013–2014. Rates of severe obesity in adolescents increased from 2.6% to 9.1% during this time period [2]. The 2016 NHANES data reports even higher rates of obesity in older adolescents, with an obesity rate of 20.2% and severe obesity rate of 9.5% in 16 to 19 year olds [1].

Children with obesity have a higher prevalence of cardiometabolic risk factors including low HDL, elevated triglycerides, elevated blood pressure, and abnormal glucose metabolism in comparison to their normal weight peers [3]. The degree of cardiometabolic risk rises with worsening severity of obesity [3,4]. Additional comorbidities associated with obesity include obstructive sleep apnea, nonalcoholic fatty liver disease, orthopedic conditions, polycystic ovarian syndrome, type 2 diabetes mellitus, and mental health concerns such as being victims of bullying [4–9]. Pediatric obesity often leads to adult obesity and shortened life-span [4,5,10]. The World Obesity Federation predicts that if current trends in obesity continue, 268 million children age 5 to 17 year old will be overweight or obese worldwide in the year 2025. Of these children, approximately 4 million will suffer from type 2

diabetes mellitus, 38 million from hepatic steatosis, 27 million from hypertension, and 12 million will have impaired glucose tolerance [11].

Recognizing the high rate of childhood obesity and the medical complications associated with this disease, it is critical to properly diagnose, treat, and prevent this condition. Obesity and excess adiposity are defined by the Body Mass Index (weight/height$^2$) [12]. In children, normal BMI values change with age and gender, so values are compared to gender- and age-specific references. For children age 2–19 years old, "overweight" is defined as having a BMI greater than or equal to 85% when plotted on the Centers for Disease Control and Prevention's gender-specific BMI-For-Age growth chart. Obesity is defined as a BMI at or above 95%. Severe obesity is defined as having a BMI at or above 120% of the 95th percentile for BMI [13,14]. For public health purposes, BMI is a reliable anthropometric estimate of adiposity [15]. However, BMI does not differentiate between Fat Mass (FM) and Free Fat Mass (FFM). Changes in diet and exercise can affect FM and FFM without necessarily affecting weight and BMI or BMI percentile [15–17]. When treating a child or adolescent, BMI percentile is an adequate screening tool to diagnosis obesity but may not be sufficient in monitoring an individual patient's treatment progress. Including body composition in the monitoring of a patient may provide a more accurate assessment of the patient's progress. Bioelectrical impedance analysis (BIA) is a convenient method to assess the body composition of children and adolescents, and has been demonstrated to be accurate relative to dual x-ray absorptiometry (DXA) [18–20]. BIA reproducibly estimates body fat percentage, making it a reliable option to track changes in body composition over the course of treatment [18]. Unlike BMI, percentile norms for body fat mass or body fat percentage by age and gender are not well established. However, general trends in change in body fat percentage with age by gender are accepted, and studies have looked at establishing percentiles and reference curves for underweight, normal, overweight, and obese [21,22]. Starting at 5 years of age, the body fat percentage increases in both boys and girls until approximately age 11 years or the onset of puberty. At that time, males will increase muscle mass with a decrease in overall body fat. In contrast, females increase body fat percentage during puberty. These changes in body composition are not accounted for by BMI. A female entering puberty with a decrease or even stabilization of body fat percentage and an increase in muscle mass, weight, and height is an example of lifestyle change in a positive direction that would not be reflected through monitoring BMI percentile alone. The purpose of this study is to determine the proportion of patients that achieve favorable changes in body composition in the absence of improvements in BMI. We categorize favorable change in body composition as a decrease in body fat percentage and an increase in skeletal muscle mass. We recognize that for females in early puberty, a stabilization of body fat percentage may be considered favorable.

## 2. Methods

The Institutional Review Board at Cincinnati Children's Hospital approved the study protocol. This was a retrospective chart review of clinical measures in children with overweight and obese weight status. Data from 52 months of clinical visits to a pediatric weight management program were extracted from electronic medical records. Height, weight, and body composition measurements were collected during clinical care. Height and weight were used to calculate BMI. BMI percentile (BMI%ile) for age and gender was determined, as well as percent of the 95th percentile for BMI (BMI%95). Bioelectrical impedance analyzers (InBody 230) were used to measure body fat percentage (PBF). Data were analyzed to determine what proportion of patients had a favorable decrease in PBF despite showing an increased or unchanged BMI.

Statistical Analysis Software (SAS$^®$) was used to clean and process data. Changes in BMI, BMI%ile, BMI%95 and PBF were grouped as follows:

BMI: $<-5$ kg/m$^2$, $-5- < -1$ kg/m$^2$, $-1 < 1$ kg/m$^2$, $1- < 5$ kg/m$^2$, $5+$ kg/m$^2$

BMI Percentile: $<-1\%$, $-1- < 0\%$, $0- < 0.4\%$, $0.4+\%$

BMI%95: $<-5\%$, $-5- < 0\%$, $0- < 5\%$, $5+\%$

PBF: $<-5\%$, $-5- < -2\%$, $-2- < 0\%$, $0- < 3\%$, $3+\%$

Mean values and mean changes in these variables were calculated. Two-way frequency tables were constructed using these groupings. Proportions of individuals falling into specific cells of these tables were noted. Paired *t*-tests were performed to determine if changes were significant. Paired *t*-tests were performed by gender for subjects whose change in BMI was positive, and separately for a change in BMI greater than 1 kg/m$^2$. Results were considered significant at a value of $p \leq 0.05$.

## 3. Results

Data were obtained for 1738 patients (941 females, 797 males), ages 4–21 years old, with at least two clinical visits. Initial age (±SD) was 12.2 ± 3.1 years. Initial BMI was 32.8 ± 7.0 kg/m$^2$. Initial BMI%ile was 98.6 ± 1.7. Initial BMI%95 was 133.2 ± 23.3. Initial PBF was 44.0 ± 6.4% (Table 1).

**Table 1.** Characteristics of participants.

| Variables | Group (n = 1738) | Males (n = 797) | Females (n = 941) |
|---|---|---|---|
| Age (year) | 12.2 ± 3.1 | 12.2 ± 3.0 | 12.2 ± 3.1 |
| BMI (kg/m$^2$) | 32.8 ± 7.0 | 32.8 ± 6.9 | 32.9 ± 7.1 |
| BMI%ile | 98.6 ± 1.7 | 98.8 ± 1.6 | 98.4 ± 1.8 |
| BMI%95 | 133.2 ± 23.3 | 135.8 ± 23.7 | 131.0 ± 22.7 |
| PBF | 44.0 ± 6.4 | 42.6 ± 6.8 | 45.2 ± 5.8 |

At follow-up, there was an overall increase in BMI (1.2 ± 3.0 kg/m$^2$, $p < 0.0001$) across the study population. However, BMI%ile, BMI%95, and PBF decreased (−0.3 ± 1.7, $p < 0.0001$; −0.8 ± 10.0, $p = 0.0006$; −0.66 ± 3.94%, $p < 0.0001$). While only 593 patients (34%) saw a decreased BMI, BMI%ile decreased in 986 patients (58%), BMI%95 decreased in 955 patients (55.8%), and PBF decreased in 928 patients (53%) (Tables 2–4).

**Table 2.** Comparison of changes in BMI to changes in BIA.

| Change in BMI | Change in Body Fat Percentage | | | | | |
|---|---|---|---|---|---|---|
| | <−5% | −5 to <−2% | −2 to <0% | 0 to <3% | 3%+ | TOTAL |
| <−5 kg/m$^2$ | 18 | 2 | 1 | 0 | 0 | 21 |
| | | | | | | 1.21% |
| −5 to <−1 kg/m$^2$ | 87 | 76 | 65 | 16 | 1 | 245 |
| | | | | | | 14.15% |
| −1 to <1 kg/m$^2$ | 45 | 126 | 269 | 231 | 16 | 687 |
| | | | | | | 39.67% |
| 1 to <5 kg/m$^2$ | 23 | 58 | 125 | 292 | 101 | 599 |
| | | | | | | 34.58% |
| 5+ kg/m$^2$ | 4 | 12 | 17 | 67 | 80 | 180 |
| | | | | | | 10.39% |
| TOTAL | 177 | 274 | 477 | 606 | 198 | 1732 |
| | 10.22% | 15.82% | 27.54% | 34.99% | 11.43% | |

**Table 3.** Comparison of Changes in BMI Percentile to Changes in BIA.

| Change in BMI Percentile | Change in Body Fat Percentage | | | | | |
|---|---|---|---|---|---|---|
| | <−5% | −5 to <−2% | −2 to <0% | 0 to <3% | 3%+ | TOTAL |
| <−1% | 84 | 49 | 25 | 14 | 1 | 173 |
| | | | | | | 10.17% |
| −1 to <0% | 70 | 170 | 289 | 248 | 36 | 813 |
| | | | | | | 47.80% |
| 0 to <0.4% | 17 | 40 | 137 | 274 | 85 | 553 |
| | | | | | | 32.51% |
| 0.4%+ | 3 | 13 | 16 | 61 | 69 | 162 |
| | | | | | | 9.51% |
| TOTAL | 174 | 272 | 467 | 597 | 191 | 1701 |
| | 10.23% | 15.99% | 27.45% | 35.10% | 11.23% | |

**Table 4.** Comparison of Changes in Percent of the 95th Percentile for BMI to Changes in BIA.

| Change in BMI%95 | Change in Body Fat Percentage | | | | | |
|---|---|---|---|---|---|---|
| | <−5% | −5 to <−2% | −2 to <0% | 0 to <3% | 3%+ | TOTAL |
| <−5% | 134 | 123 | 125 | 54 | 4 | 440 |
| | | | | | | 25.72% |
| −5 to <0% | 27 | 99 | 196 | 173 | 20 | 515 |
| | | | | | | 30.10% |
| 0 to <5% | 9 | 34 | 114 | 215 | 43 | 415 |
| | | | | | | 24.25% |
| 5%+ | 6 | 16 | 35 | 155 | 129 | 341 |
| | | | | | | 19.93% |
| TOTAL | 176 | 272 | 470 | 597 | 196 | 1711 |
| | 10.29% | 15.90% | 27.47% | 34.89% | 11.46% | |

Both males and females demonstrated a significant increase in BMI, yet both had significant decreases in BMI%ile. Only males had significant decreases in BMI%95 and PBF, while those measures remained stable in females (Table 5).

**Table 5.** Mean Changes Between Visits by Gender.

| Gender | Variable | Mean Change | Significance |
|---|---|---|---|
| Female | BMI | 1.47 kg/m$^2$ | $p < 0.0001$ |
| | BMI%ile | −0.25 | $p < 0.0001$ |
| | BMI%95 | −0.06 | $p = 0.83$ |
| | Skeletal Muscle | 2.25 kg | $p < 0.0001$ |
| | Fat Mass | 3.57 kg | $p < 0.0001$ |
| | % Body Fat | 0.07% | $p = 0.47$ |
| Male | BMI | 0.88 kg/m$^2$ | $p < 0.0001$ |
| | BMI%ile | −0.38 | $p < 0.0001$ |
| | BMI%95 | −1.73 | $p < 0.0001$ |
| | Skeletal Muscle | 3.13 kg | $p < 0.0001$ |
| | Fat Mass | 2.04 kg | $p < 0.0001$ |
| | % Body Fat | −1.53% | $p < 0.0001$ |

BMI increased or remained unchanged in 1148 patients (66%). By gender, 656 females (70%) and 492 males (62%) saw a stable or increased BMI. In those patients whose BMI increased or remained unchanged, overall BMI%ile, BMI%95 and PBF increased (0.15 ± 0.89, *p* < 0.001; 3.4 ± 7.6, *p* < 0.0001; 0.55 ± 3.15%, *p* < 0.0001). In males, the increase in PBF was small (0.06 ± 3.34, *p* = 0.7). Of the 779 patients whose BMI increased by 1.0 kg/m$^2$ or more, 239 (30.1%) still saw a decrease in PBF (Table 2).

BMI%ile increased or remained unchanged in 715 patients with available data (42%). Of these patients, 226 (31.6%) had a decrease in PBF (Table 3). BMI%95 increased or remained unchanged in 756 patients with available data (44%). Of these, 214 (28%) had a decrease in PBF (Table 4).

## 4. Discussion

The children and adolescents reviewed in this study had, at baseline, a mean BMI%95 of 133.2%, which is classified as severe obesity/Class 2 Obesity, and a body fat percentage of 44%, which is in the obese range for every age and gender [21,22]. To decrease the risk of long-term health consequences of obesity, these patients participated in our pediatric weight management program. Accurately monitoring their progress in achieving improved health is critical in evaluating their treatment, as well as motivating them in continued efforts in lifestyle modification. At follow up, there was an overall increase in BMI. BMI increases with age throughout childhood and adolescence for males and females. For this reason, age- and gender-based BMI percentiles are used to clinically track the balance of weight and height in youth. BMI%95 is more accurate in assessing change in the upper extreme of the BMI percentile curve. The patients in this study had an overall improvement in BMI%ile and BMI%95 at −0.3% and −0.8% respectively. Additionally, the patients had an overall decrease in PBF of −0.66%. As children and adolescents focus on lifestyle change with healthy eating and exercise, as recommended by pediatric weight management programs, they typically gain skeletal muscle mass. This was seen in both the male and female patients reviewed in our study (3.13 kg and 2.25 kg respectively). As a group, the patients in this study would be considered to have made progress in lifestyle modification, thereby positively impacting their health. Looking at the patients by gender, males showed significant decrease in BMI%ile, BMI%95, and PBF, with an increase in muscle mass, while females showed decrease in BMI%ile but stable BMI%95 and PBF with increased muscle mass. For the male patients overall, each of these four markers would indicate success if used to monitor treatment progress alone. That is not the case in the female patients. When looking at the information in combination, however, a decrease in BMI% and an increase in skeletal muscle mass with a stable body fat percentage may indicate success, particularly in females for whom the trend is increasing body fat percentage. Although body composition in addition to BMI percentiles are helpful in evaluating the overall success of a program, the combined information is particularly helpful in tracking the success of an individual participating in a weight management program. Having information regarding body composition as well as BMI%ile allows for more precise evaluations of the success of treatment in individual patients participating in pediatric weight management programs.

Looking at our data, some patients with increasing muscle mass and decreasing body fat percentage had stable or increasing BMIs, BMI%ile, and BMI%95, despite positive lifestyle change. Overall, patients with stable or increasing BMIs did not show improvement in body fat percentage. However, in select patients, although the BMI remained unchanged or increased, the patient's body fat percentage decreased. Among patients whose BMI increased by 1.0 kg/m$^2$ or more, 30% demonstrated an improvement in body composition (skeletal muscle mass and body fat percentage). Similarly, 31.6% of patients with stable or increasing BMI%ile and 28% with stable or increasing BMI%95 showed improvement in body fat percentage. This additional information is helpful in tracking the progress of individuals participating in a weight management program. Body composition is an important measure of success for a subset of patients who otherwise may believe that their efforts in lifestyle change had not been beneficial. Noting improvement in body fat percentage and skeletal muscle mass may act as a motivator for patients to continue with lifestyle changes. Our results suggest that including body fat percentage as a measure of success in evaluating the progress of patients

participating in a pediatric weight management program is appropriate and may more accurately track success than change in BMI, BMI%95, or BMI percentile alone. Further research is needed in this area to confirm our results and to evaluate the utility of body composition analysis by different age groups, class of obesity, gender, and how these variations change over time.

**Author Contributions:** K.S.; R.S.; P.K.; A.G. have each contributed to the conception of the study, analysis and interpretation of the data, drafting and revision of the manuscript, and have approved the submitted version. Each author agrees to be personally accountable for the author's own contributions and for ensuring that questions related to the accuracy or integrity of any part of the work, even ones in which the author was not personally involved, are appropriately investigated, resolved, and documented in the literature. All authors have read and agreed to the published version of the manuscript.

**Funding:** This research received no external funding.

**Conflicts of Interest:** The authors declare no conflict of interest.

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
