# Peer review of "Body Composition versus BMI as Measures of Success in a Clinical Pediatric Weight Management Program"

_reports, doi:10.3390/reports3040032_

Round 1

Reviewer 1 Report

The aim of the study is clear.

It might be more precise to transform BMI values to SDS, rather than comparing changes in BMI percentiles. You could ask for a statistician opinion.

Discussion is a bit short. You should compare your results with similar published studies.

Author Response

Dear Ms. Wang, Editorial Staff, and Reviewers,

Thank you for your thoughtful feedback. We appreciate the time that you have spent in reviewing our manuscript.  The reviewer comments and our responses are: 

  1. It might be more precise to transform BMI values to SDS, rather than comparing changes in BMI percentiles. You could ask for a statistician opinion.

Because BMI norms change with age and gender, BMI percentiles are preferred value for evaluating the pediatric population. BMI percentiles allow for a uniform metric across all ages.  For the upper extreme end of the BMI percentile, percent of the 95th percentile for BMI is considered the standard in identifying and assessing change in patients with severe (class 2 and 3) obesity. I tried to clarify that within the text in both the discussion and the introduction. Introduction page 2, line 51, 54, 55; Discussion page 5 line 127-131

Our statistician did run the numbers with BMI z-scores. They correlated well with the BMI%95. I am happy to supply you with that analysis if it would be helpful. We chose not to include it in the manuscript since change in  BMI%ile and BMI%95 are the standard both clinically and in research currently.

  1. Discussion is a bit short. You should compare your results with similar published studies.

To the best of our knowledge, there are not similar studies for comparison. (Page 5 line 167). We did work to make the discussion more comprehensive by adding more discussion of the results. Page 5: added line 122 to 148, removed line 149 to the word study on 151. Added “Looking at our data,” (line 151). Made small changes for a more consistent message: line 153, line 158, line 159, line 160, line 162, line 163

Reviewer 2 Report

The article entitled „Body composition versus BMI as measures of success in a clinical Pediatric weight management program” could be very interesting and valuable paper for a research and clinical point of view. It underlays very important issue of therapeutic approach to children with obesity managed in weight control programs.

However, in my opinion, the results should be better presented. As BMI also PBF change with age, especially that part of the participants were managed during puberty. The PBF should also be presented in relation to age and sex (as BMI – ex. as percentiles).  The discussion section needs major improvement – in the present version there is no debate of the results at all.  

In my opinion paper needs major changes before publication.

Author Response

Dear Ms. Wang, Editorial Staff, and Reviewers,

Thank you for your thoughtful feedback. We appreciate the time that you have spent in reviewing our manuscript. The reviewer comments and are responses are:

  1. …the results should be better presented. As BMI also PBF change with age, especially that part of the participants were managed during puberty. The PBF should also be presented in relation to age and sex (as BMI – ex. As percentiles).

Unfortunately, there are not widely accepted percentile norms for PBF in the pediatric population. We now  explain this more clearly in the introduction page 2 line 61-70.

References # 21 and 22 were added, page 7 line 226-229

  1. The discussion section needs major improvement – in the present version there is no debate of results at all.

To the best of our knowledge, there are not similar studies for comparison. (Page 5 line 167). We did work to make the discussion more comprehensive by adding more discussion of the results. Page 5: added line 122 to 148, removed line 149 to the word study on 151. Added “Looking at our data,” (line 151). Made small changes for a more consistent message: line 153, line 158, line 159, line 160, line 162, line 163

Round 2

Reviewer 2 Report

Thank you for the aswers to my comments and for the imrovements you have made